# Exploring the Value of *BRD9* as a Biomarker, Therapeutic Target and Co-Target in Prostate Cancer

**DOI:** 10.3390/biom11121794

**Published:** 2021-11-30

**Authors:** Nafisa Barma, Timothy C. Stone, Lina Maria Carmona Echeverria, Susan Heavey

**Affiliations:** 1Division of Surgery and Interventional Sciences, University College London, London WC1E 6BT, UK; nafisa.barma.18@ucl.ac.uk (N.B.); rmhkton@ucl.ac.uk (T.C.S.); linacarmona@nhs.net (L.M.C.E.); 2Department of Urology, University College London Hospitals NHS Foundation Trust, London WC1H 8NJ, UK

**Keywords:** BRD9, prostate cancer, SWI/SNF, targeted therapy

## Abstract

Background and aims: Despite recent advances in advanced prostate cancer treatments, clinical biomarkers or treatments for men with such cancers are imperfect. Targeted therapies have shown promise, but there remain fewer actionable targets in prostate cancer than in other cancers. This work aims to characterise *BRD9*, currently understudied in prostate cancer, and investigate its co-expression with other genes to assess its potential as a biomarker and therapeutic target in human prostate cancer. Materials and methods: Omics data from a total of 2053 prostate cancer patients across 11 independent datasets were accessed via Cancertool and cBioPortal. mRNA M.expression and co-expression, mutations, amplifications, and deletions were assessed with respect to key clinical parameters including survival, Gleason grade, stage, progression, and treatment. Network and pathway analysis was carried out using Genemania, and heatmaps were constructed using Morpheus. Results: *BRD9* is overexpressed in prostate cancer patients, especially those with metastatic disease. *BRD9* expression did not differ in patients treated with second generation antiandrogens versus those who were not. *BRD9* is co-expressed with many genes in the SWI/SNF and BET complexes, as well as those in common signalling pathways in prostate cancer. Summary and conclusions: *BRD9* has potential as a diagnostic and prognostic biomarker in prostate cancer. *BRD9* also shows promise as a therapeutic target, particularly in advanced prostate cancer, and as a co-target alongside other genes in the SWI/SNF and BET complexes, and those in common prostate cancer signalling pathways. These promising results highlight the need for wider experimental inhibition and co-targeted inhibition of *BRD9* in vitro and in vivo, to build on the limited inhibition data available.

## 1. Introduction

### 1.1. Prostate Cancer and Its Biomarkers

Prostate cancer (PCa) is the second most common cancer in men and the most commonly diagnosed in the UK, representing 7.1% of new cancers and 3.8% of cancer deaths worldwide [1,2,3,4].

Prostate-specific antigen (PSA) is a protein typically produced in the prostate and used as a biomarker to prompt further investigation (e.g., biopsy) for PCa and its recurrence) [5]. PSA does not provide any information about the aggressivity and stage of any potential cancer, and testing is often associated with overdiagnosis and overtreatment, with a recent meta-analysis finding that screening did not impact overall mortality, despite men who were screened having a higher incidence of PCa [6]. There are efforts to improve the diagnosis and discovery of dx biomarkers, including the United States Food and Drug Administration (FDA)-approved Progensa assay (PCA3 (PCa gene 3) and PSA) and STHLM3 (Stockholm-3), which is currently being validated [7,8,9]. However, these are imperfect, and more sensitive and specific biomarkers are needed both for detecting PCa and driving treatment [10].

### 1.2. Current Therapies for PCa

Though PCa survival is relatively high (85% five-year survival), this is much reduced (50%) for stage four PCa [11]. Treatments are often invasive with side effects for radical prostatectomy (surgical removal of the prostate) and radiotherapy, including urinary incontinence and erectile dysfunction [12,13]. Therapies for PCa range from hormonal therapies (including androgen deprivation therapy (ADT)) to chemotherapy and recently approved targeted therapies. Targeted therapies aim to inhibit specific proteins and are have shown huge promise as a future area of therapeutics for PCa, with the recent FDA approval of the poly-ADP ribose polymerase inhibitors (PARPi) olaparib and rucaparib for metastatic castration-resistant prostate cancer (mCRPC) [14].

With poor outcomes for patients with advanced stage cancer and the development of resistance to existing therapies, *BRD9* may have utility along the PCa clinical pathway as a biomarker and a therapeutic target. New targeted therapies, such as those targeting *BRD9*, with less toxic and life-altering effects would be of great benefit in treating patients with PCa. 

### 1.3. The Relationship between BRD9 and SWI/SNF and BET Complexes

*BRD9* is a genetic subunit of the non-canonical barrier-to-autointegration factor (ncBAF), also termed GBAF. This is one of the subunits composing the switch/sucrose non-fermentable (SWI/SNF) complex along with canonical barrier-to-autointegration factor (cBAF) and Polybromo-barrier-to-autointegration factor (PBAF). These complexes are involved in chromatin remodelling and cancer, containing both oncogenes and tumour suppressor genes [15]. Similarly, the Bromodomain and extra-terminal domain (BET) complex is involved in regulating transcription by RNA polymerase II [16]. This represents a possible therapeutic target in PCa with the selective small-molecule BET inhibitors JQ1 and I-BET preventing growth of androgen receptor (AR) positive castrate-resistant prostate cancer (CRPC) cell lines [17]. Both BET complex and *BRD9* inhibition have also been shown to have similar effects on gene expression [18]. BRD9 has been shown to associate with BRD2 and BRD4 proteins in the BET complex; Alpsoy et al. showed BRD9 immunoprecipitated BRD2 and BRD4, and when either BRD9 or BET complex proteins were inhibited, their interaction was reduced [19].

### 1.4. Common Signalling Pathways in PCa

Pathways involved in driving PCa proliferation involve the JAK-STAT (janus kinase; signal transducer and activator of transcription), MAPK (mitogen-activated protein kinase), and PI3K-AKT-mTOR (phosphoinositide 3-kinase; Protein Kinase B; mammalian target of rapamycin) pathways. 

The JAK-STAT pathway is regulated by *SOCS* (suppressor of cytokine signalling) genes and is involved in regulation of “cell growth, differentiation, proliferation, invasion, survival, and inflammation” [20]. Its persistent activation in PCa can result in progression, rendering it a worthwhile therapeutic target, with one study finding *STAT5A/B* gene amplification and increased protein expression in PCa, and to a greater extent, in higher grade, castration-resistant and metastatic disease [20,21]. The MAPK pathway is also involved in cell regulation and represents a possible therapeutic target in PCa, with one study finding patients with increased nuclear MAPK protein expression more likely to develop CRPC, earlier biochemical relapse, and reduced survival [22,23]. Another study found that 32% of patients with mCRPC have frequent amplifications in MAPK pathway-involved genes, and that this pathway is a promising target for mCRPC, with extracellular receptor kinase 1 (ERK1) inhibitor trametinib (currently approved for melanoma) being investigated for mCRPC in a phase II trial [24]. PI3K-AKT-mTOR is another often-dysregulated signalling pathway, especially in CRPC [25]. Though no drugs targeting this pathway have been approved in PCa, the AKT-inhibitor ipatasertib performed well in combination with abiraterone in a phase Ib/II study, and the IPATential150 phase III trial is underway [26,27]. Initial data shows a higher radiographic progression-free survival in the ipatasertib compared to the control group (18.5 months vs. 16.5 months), though it is too early for overall survival (OS) data and conclusions to be drawn [28]. Transmembrane protease, serine 2-erythroblast transformation-specific (*TMPRSS2-ETS*) genetic fusion, is a molecular subtype (ETS-related gene (*ERG*) positive) of PCa associated with cancer invasiveness found in 50% of PCa tumours [29,30]. Agents such as the small-molecule selective *ERG* inhibitor ERGi-USU (1-[2-Thiazolylazo]-2-naphthol) are being developed to target this fusion [31].

These pathways may represent opportunities for co-targeting with *BRD9* as well as being targets themselves.

### 1.5. BRD9 and PCa

BRD9 has been generating interest in the cancer research community in recent years, with data emerging in cancers such as squamous cell lung cancer, synovial sarcoma, and malignant rhabdoid tumours, where it has been shown to induce cancer hallmarks, correlate with aggression, and crucially, inhibition has led to anti-cancer effects [32,33,34,35]. However, despite this work in other cancers, *BRD9* has not been well studied in PCa to date, with only one paper published in December 2020, which directly focused on *BRD9* expression and inhibition in the disease [19,32,33,36]. This paper showed that *BRD9* inhibition and knockdown have overlapping effects, reducing AR-positive cell line growth both in vivo and in vitro. The authors also found that BRD9 interacts with the AR in PCa cell lines, even those resistant to androgen deprivation and inhibition. They also provided cell-line evidence that BRD9 and the BET complex associate with each other and interact given they have overlapping transcriptional targets. The authors previously investigated the SWI/SNF complex and how its components *GLTSCR1* and *GLTSCR1-L* are involved in chromatin remodelling and along with *BRD9* form a unique subcomplex however they did not focus on *BRD9* and PCa here [37].

Despite publishing the first paper on BRD9 in PCa, the authors did not begin with an unbiased characterisation of this gene in human PCa using extensive existing publicly available data. Instead, they focussed almost exclusively on CRPC (developed by 10–20% PCa patients within 5 years) and worked with cell lines [38]; much remains to be answered regarding BRD9’s behaviour across the clinical pathway. However, the authors’ findings were encouraging, warranting further investigation, which could lead to clinical trials if future results continue to have promise.

Here, we set out to determine whether BRD9 represents a potential biomarker, monotherapeutic target, and co-target alongside other targeted therapies in PCa.

## 2. Materials and Methods

### 2.1. Cancertool, cBioPortal and Genemania

We used publicly available data to investigate *BRD9* as a biomarker and therapeutic target and co-target in PCa. With regard to *BRD9* as a biomarker, we compared *BRD9* mRNA expression in samples from patients with and without PCa, those with different grades and stages of PCa, and patients who received and responded to treatment. When investigating *BRD9* as a therapeutic target and co-target in PCa, we looked at how *BRD9* mRNA expression was correlated with AR messenger ribonucleic acid (mRNA) expression and other genes involved in common PCa signalling pathways as well as the BET complex, as well as interactions between *BRD9* and some of these genes. We then graphed and analysed these data.

Cancertool was used to access *BRD9* mRNA expression in PCa cohorts as well as gene mRNA correlations in PCa. It is a publicly available interface that generates graphical visualisations of data, performs some statistical analyses, and allows users to download raw data for processing and analysis [39].

The cBioPortal for Cancer Genomics was also used to access PCa cohort data. This platform integrates various genomic data, allowing users to investigate, view, and download data relating to gene expression, mutation data (for some cohorts), and clinical attributes [40,41].

Genemania performs network analyses between inputted genes and presents relationships as a colour-coded network [42]. 

The following cohorts (cancer omics datasets) were accessed via Cancertool and cBioPortal (Table 1).

### 2.2. Gene Correlations and Co-Expression Analysis

Gene correlations were investigated by downloading raw data from Cancertool and cBioPortal and processing the data using Microsoft Excel. Datasets accessed are summarised above in Table 1. Following this, data were graphed, and statistical analyses carried out using GraphPad Prism version 9.1.0. 

Co-expression between *BRD9* and genes involved in signalling pathways and complexes of interest were investigated using Genemania and Cancertool. Spearman’s rank-order correlation coefficient (R) and its statistical significance were calculated in Cancertool. Correlation coefficient data was then processed using Microsoft Excel, and heatmaps were created using Morpheus to visually represent the correlation between *BRD9* expression and expression of other related genes of interest [54]. Once generated, the heatmaps were examined and *p* values were manually indicated by adding ‘*’ to any pixel whose correlation was significant, with significance determined at α = 0.05.

### 2.3. Comparing Continuous Data

Before comparisons were performed, data were graphed and tested for normality using Shapiro–Wilk and Kolmogorov–Smirnov tests. All data were unpaired. When comparing two groups, normally distributed data were analysed using Welch’s *t*-test, as group standard deviations differed. Non-normally distributed data were analysed using a Mann–Whitney U test. Where more than two groups were to be compared, normally distributed data were analysed using the Brown–Forsythe and Welch’s analysis of variance (ANOVA) test as group standard deviations differed. Dunnett’s T3 test was then performed as a post hoc multiple comparisons test. Non-normally distributed data were analysed using the Kruskal–Wallis test. Dunn’s test was then performed as a post hoc multiple comparisons test.

Following statistical analysis, data were graphed using violin plots, as these better illustrate data distribution and density than bar graphs and box and whisker plots [55].

### 2.4. Correlations

Spearman’s R correlations could be classified as very weak, weak, moderate, strong or very strong (Table 2) [56].

### 2.5. Survival Analysis

Disease-free survival (DFS) was illustrated using Kaplan–Meier survival curves. Patients were grouped by quartile expression of *BRD9*. A Mantel–Cox test was performed to compare the differences between curves and a cox regression model was used to calculate hazard ratio (HR) between the first and fourth quartiles.

### 2.6. Comparing Mutation Distribution

Where somatic mutation data (amplification, gain of function, missense, shallow deletion, and deep deletions) was available, this was summarised and graphed using stacked bar charts. As categorical variables investigated had small sample sizes and expected frequencies < 1 for some cells, Fisher’s exact test was required to analyse mutation distribution. This test was implemented in R. In some instances, modified estimates of the *p*-value were necessary, as certain tables were too computationally intensive to calculate. The approximation was calculated via a Monte Carlo simulation (number of samples = 1,000,000). In every instance, the *p*-values indicated that the data were considerably below the threshold for significance (*p* < 10^−5^).

## 3. Results

### 3.1. BRD9 Has Potential as a Diagnostic and Prognostic Biomarker in PCa

*BRD9* is overexpressed in human prostate tumour tissue, playing a role as a diagnostic biomarker. It may also be overexpressed in more aggressive PCa, and this overexpression may correlate with reduced survival in patients. Comparison of *BRD9* expression in the Grasso cohort revealed that *BRD9* is overexpressed in PCa patients (*p* = 0.0462) (Figure 1A). In the Taylor cohort, there was no significant difference in *BRD9* expression (p = 0.6726); however, this was positively skewed (Figure 1B). The Varambally cohort also showed greater *BRD9* expression in PCa (*p* = 0.0053) compared to benign samples from the same prostate (Figure 1C). Regarding Gleason grade and *BRD9* mRNA expression, using Gleason grade 6 as a baseline, there was no significant change in *BRD9* expression level as Gleason grade increased in the Glinsky cohort (*p* > 0.05) (Figure 1D). In the Taylor cohort, there was only a significant change (decrease) in *BRD9* expression at Gleason grade 8 (*p* = 0.0100) (Figure 1E). In the TCGA cohort, there was no significant difference between Gleason grade and *BRD9* expression (*p* > 0.05) (Figure 1F). *BRD9* mutations were more prevalent in patients with higher graded disease (*p* < 0.0001) (Figure 1G). In terms of *BRD9* expression in normal prostate, primary, and metastatic PCa, in the Grasso cohort, there was no significant difference (*p* = 0.9268) in *BRD9* expression between normal prostate and primary PCa; however, there was a significant increase in *BRD9* expression in the metastatic PCa group, both when compared to benign prostate (*p* = 0.0025) and primary PCa (*p* = 0.0016) groups (Figure 1H). Similarly, in the Taylor cohort, there was only a significant increase in *BRD9* expression between the normal prostate and metastatic (*p* = 0.0152), and primary and metastatic groups (*p* = 0.0017) (Figure 1I). The Varambally cohort only showed a significant change in *BRD9* expression between the normal and metastatic groups (*p* = 0.0270) (Figure 1J). In the Glinsky cohort, patients with higher *BRD9* expression had significantly higher DFS (*p* = 0.0190) (Figure 1K). In the Taylor cohort, there was no significant difference (*p* = 0.8532) in DFS between patients with the lowest and highest *BRD9* expression (Figure 1L). In the TCGA cohort, patients with higher *BRD9* levels had significantly lower DFS than those with lower levels (*p* = 0.0015) (Figure 1M).

### 3.2. BRD9 Is Not Overexpressed in Patients with Advanced Stage PCa

*BRD9* expression in advanced stage PCa was assessed. In all cohorts investigated, there was no significant change in *BRD9* expression in more advanced stage disease (*p* > 0.05) (Appendix A). In the TCGA cohort, *BRD9* mutations were more prevalent in patients with more advanced disease (*p* < 0.0001) (Appendix A); however, this did not appear to be the case in the Ren cohort (*p* = 0.7062) (Appendix A).

### 3.3. BRD9 Does Not Appear to Play a Role as a Predictive Biomarker in PCa

*BRD9* expression levels were assessed in patient responses to therapy and in patients undergoing various therapeutic regimens. Data regarding this were lacking in most cohorts. There was no significant difference in *BRD9* expression between primary therapy outcomes in the TCGA cohort (*p* > 0.05) (Figure 2A). *BRD9* mutations were less prevalent in patients with a complete response to primary therapy in this cohort (*p* < 0.0001) (Figure 2B). There was no significant difference in *BRD9* expression in patients indicated for adjuvant radiotherapy in the TCGA cohort (*p* = 0.4795) (Figure 2C), though *BRD9* mutations were more prevalent in patients indicated for adjuvant radiotherapy here (*p* < 0.0001) (Figure 2D). There was no significant difference in *BRD9* expression in patients who received chemotherapy in the Kumar cohort (*p* = 0.4564) (Figure 2E), though *BRD9* mutations were more prevalent in patients who did not receive chemotherapy (*p* < 0.0001) (Figure 2F). There was no significant difference in *BRD9* expression between different therapy regimens in the Abida cohort (*p* > 0.05) (Figure 2G). *BRD9* mutations appeared to be similarly prevalent across various regimens (*p* < 0.0001) (Figure 2H).

### 3.4. BRD9 May Play a Role as a Therapeutic Target in CRPC

*BRD9* and AR mRNA expression correlation and *BRD9* expression in patients receiving second generation antiandrogens (abiraterone and enzalutamide) was investigated. The TCGA cohort demonstrated a moderate negative correlation between *BRD9* and AR expression in PCa patients (R = −0.4117, *p* < 0.0001) (Figure 3A). There was no significant difference in *BRD9* expression and abiraterone and enzalutamide exposure status in the Abida and Dan cohorts (*p* > 0.05) (Figure 3B,D); however, *BRD9* mutations were more prevalent in patients treated with these drugs (*p* < 0.0001) (Figure 3C,E)

### 3.5. BRD9 Correlates with Genes in the SWI/SNF and BET Complexes

*BRD9* expression and its correlation with other genes in the SWI/SNF and BET complexes was investigated. *BRD9* expression was positively correlated with most genes in the SWI/SNF complex (Figure 4A–C): it was negatively correlated with *ACTB*, *BCL7A*, *BCL7B*, *BCL7C*, *SMARCA4*, *SMARCC2*, *SMARCD1*, *SMARCD2*, *SMARCD3*, *GLTSCR1*, *DPF1*, *DPF2*, *SMARCB1*, and *SMARCE1* and negatively correlated with *SMARCA2* and *ARID2* at a significant level (*p* > 0.05). The cohorts disagreed on whether *BRD9* expression was positively or negatively correlated with other genes in the SWI/SNF complex at a significant level (*p* > 0.05) (Appendix A). *BRD9* was co-expressed and co-localised and physically interacted with *SMARCD1*. It also physically interacted with *SS18* and shared a domain with *BRD7* and *PBRM1*. *BRD9* genetically interacted with DPF2 and was co-expressed with *SMARCC1*, *SMARCC2*, *BCL7B*, and *BCL7C* (Appendix A). *BRD9* was positively correlated with most genes in the BET complex (Figure 4D): it positively correlated with *BRD2*, *BRD3*, and *BRD4* at a significant level in the Grasso, Taylor, and Varambally cohorts (*p* > 0.05). *BRD9* was negatively correlated with *BRDT* at a significant level in the Taylor cohort—though it was positively correlated with *BRDT* in the other cohorts, the correlation was not significant (*p* > 0.05) (Appendix A). *BRD9* has shared protein domains with genes found in the BET complex, physically interacted with BRD4, and was co-expressed with *BRD2* (Appendix A).

### 3.6. BRD9 Correlates with Genes Involved in Common PCa Proliferation-Driving Pathways

The correlation of *BRD9* with known genes involved in the JAK-STAT, MAPK, and PI3K-AKT-mTOR pathways was investigated (Figure 5). We found that *BRD9* is negatively correlated with *JAK2* and positively with *TYK2*. *BRD9* was also positively correlated with *STAT2*, *STAT4*, *STAT5A*, *STAT5B*, and *STAT6* as well as *PIM1*, *PIM2*, and *PIM3* at a significant level (*p* > 0.05). The cohorts disagreed on its correlation with *BCL2* and *MYC*: Taylor demonstrated a significant negative correlation between *BRD9* and *BCL2*; however, Varambally demonstrated the opposite (*p* > 0.05). Taylor demonstrated a positive correlation between *BRD9* and *MYC*, whereas the Varambally cohort demonstrated the opposite. *BRD9* was positively correlated with *SOCS1* and negatively correlated with *SOCS2*, *SOCS5*, and *SOCS6*. It appeared to be positively correlated with *SOCS3* and negatively correlated with *SOCS4*. The cohorts disagreed on whether it was positively or negatively correlated with *SOCS7*—though Grasso and Varambally found significant positive correlations between *BRD9* and *SOCS7* expression, TCGA data showed a significant negative correlation between these genes (*p* > 0.05) (Appendix A). *BRD9* was co-expressed with *TYK2*, *STAT4*, and *SOCS7* (Appendix A). In all cohorts investigated, there was a positive correlation between *BRD9* expression and *HRAS* expression, though this was only significant in the Grasso, Taylor, and TCGA cohorts (*p* > 0.05). The cohorts disagreed on whether *BRD9* is correlated with *KRAS* and *NRAS* positively or negatively at a significant level (*p* > 0.05). *BRD9* was positively correlated with *ARAF*, *BRAF* and *RAF1* at a significant level in most of cohorts; however, in the TCGA cohort the correlations are negative and significant (*p* > 0.05). *BRD9* was negatively correlated with *MAP2K1* and positively correlated with *MAP2K2*. *BRD9* was positively correlated with *MAPK3* and negatively correlated with *MAPK1* (Appendix A). *BRD9* was co-expressed with *HRAS*, *RAF1*, and *ARAF* (Appendix A). *BRD9* was negatively correlated with *PIK3CA*, *PIK3CB*, *PIK3CG*, *PIK3R1*, *PIK3R3*, *PIK3R4*, *PIK3C2A*, *PIK3C2G*, and *PIK3C3* (*p* > 0.05). It was positively correlated with *PIK3R2*. *BRD9* was negatively correlated with PTEN at a significant level in the Grasso, Taylor, and TCGA cohorts (*p* > 0.05). *BRD9* was positively correlated with *PDK1* at a significant level in the Grasso and Taylor cohorts (*p* > 0.05). *BRD9* was positively correlated with *AKT1* and *AKT2*; however, it was negatively correlated with *AKT3* (*p* > 0.05). In terms of the *TOR* complexes, *BRD9* was positively correlated with *AKT1S1*, *RPTOR*, *MLST8*, *PRR5*, and *MAPKAP1* (Appendix A). *BRD9* was co-expressed with *PI3K3C2B* and *AKT2* (Appendix A).

### 3.7. BRD9 Could Play a Role as a Therapeutic Co-Target Alongside ERG

The potential *BRD9* as a co-target with *ERG* was investigated. Though not directly related to *ERG*, *BRD9* and ERG were indirectly connected via co-expression, physical, and genetic interactions (Figure 6A). PCa is either *ERG* positive or negative; *TMPRSS2* and *ERG* genes fuse, increasing the expression of *ERG* in roughly half of cases. This is shown by the two distinct groups in the scatterplot, which also shows that *BRD9* and *ERG* expression are positively correlated in the TCGA cohort (Figure 6B). *BRD9* is overexpressed in *ERG* positive PCa in the Barbieri (*p* = 0.0024) and Gerhauser (*p* = 0.0054) cohorts; however, there was no significant difference in *BRD9* expression in the Taylor (*p* = 0.1790) and Abida (*p* = 0.1304) cohorts (Figure 6C–F). *BRD9* mutations were more prevalent in the *ERG* negative group in the Abida cohort (*p* < 0.0001) (Figure 6G).

## 4. Discussion

Though BRD9 is understudied in PCa, growing interest in the community has recently pointed to a potential role in other cancers. Here, we set out to characterise *BRD9* expression and mutations in depth within 11 independent PCa cohorts to identify whether *BRD9* has potential as a biomarker and therapeutic target or co-target in PCa.

### 4.1. BRD9 as a Biomarker

SWI/SNF subunits are the most commonly mutated chromatin-regulatory complexes in human cancer, mutated in 19.6% cancers, with subunits *ARID1A*, *PBRM1*, *SMARCA4*, and *ARID2* already part of routine cancer diagnostics [15,58]. The literature suggests that *BRD9* plays an oncogenic role in many cancers with promise both as a diagnostic and prognostic biomarker [32,33,36]. The only paper on *BRD9* and PCa used the TCGA evidence to show that *BRD9* worsened DFS but did not investigate *BRD9* as a biomarker in PCa [19].

Our findings show that *BRD9* is overexpressed in PCa and therefore may play a role as a diagnostic biomarker in this disease. While *BRD9* could represent a good diagnostic biomarker in PCa, it may not necessarily mean that this is needed, with PSA being widely used. Biomarkers that more accurately predict aggressive disease are more needed, so we went on to identify whether *BRD9* could represent a more clinically useful biomarker here. Currently, researchers are investigating panels of genes rather than individual genes in PCa diagnostics [59]. *BRD9* could be a useful panel component, given most genes composing a biomarker panel would not exhibit as strong correlations with disease status as *BRD9* exhibits. Though *BRD9* expression does not change with increases in Gleason grade and tumour stage, its expression was higher in metastatic PCa when compared to cancer-free patients and patients with local disease, suggesting that *BRD9* could play a role as a prognostic biomarker and therapeutic target, especially in metastatic PCa. As with any study of historic data, it should be noted that analysis methods have changed over time here. For Gleason scoring, both pathological methods and WHO classifications have changed during the collection of these data, which could affect the conclusions here, and is an unavoidable limitation in the use of publicly available retrospective cohorts. It has been shown that BRD9 is required for cell growth, and its degradation prevents synovial sarcoma tumour progression [32,33]. Similar findings have been reported in hepatocellular carcinoma (HCC), with BRD9 overexpressed in HCC patients as well as promoting cell growth and metastasis, and its depletion and inhibition reducing these effects in HCC cells [36]. Data on DFS and *BRD9* expression was conflicting; however, the widely studied TCGA cohort with the largest sample size found patients with higher *BRD9* expression to have worse prognoses. Similarly, higher BRD9 expression reduces OS and DFS in HCC [36]. In the TCGA cohort, there is evidence suggesting that *BRD9* is more likely to be mutated in higher Gleason grades and more advanced cancer, and screening these groups and targeting BRD9 in patients with mutations may be a viable therapeutic strategy. Though these data provide limited evidence that BRD9 plays a role as a predictive biomarker in PCa, it suggests that mutations are more prevalent in patients indicated for radiotherapy. Further work should be carried out to determine if this means *BRD9* expression correlates with some other factor that leads clinicians to indicate patients for radiotherapy. Targeting BRD9 may be synergistic with adjuvant radiotherapy, with one study finding BRD9 sensitised ovarian cancer cell lines to radiotherapy [60].

### 4.2. BRD9 and CRPC

Alpsoy et al.’s paper on BRD9 in PCa suggests that BRD9 could play a role in CRPC with BRD9 knockdown, reducing the viability of AR-positive cell lines (including castration-resistant cell lines) [19]. Here, this cell line data was further characterised in 2053 PCa patients and did not suggest as promising a role, given *BRD9* expression was similar in patients treated with and without second-generation antiandrogens. However, the decision to treat or not treat patients is made in the absence of knowledge of *BRD9* expression, and naturally, the optimal experiment would be to compare *BRD9* expression with response to antiandrogens in all patients—data that are not currently available within these datasets. While the cell line data is promising, cell lines cannot create a biomimetic environment and can become contaminated or subject to genetic drift [61,62,63]. We look forward to clinical outcome data becoming available, allowing us to further investigate these findings in more relevant settings.

*BRD9* expression is moderately negatively correlated with AR expression in the TCGA cohort. Though this cohort was treatment-naïve, the correlation suggests that *BRD9* could be a viable target in patients with AR-negative disease. After PCa differentiates into CRPC (stops responding to ADT) and is treated with second generation antiandrogens, it may stop responding to these also and then de-differentiate into AR-negative disease [64]. Though not in line with existing literature, if successful in AR-negative PCa, targeting *BRD9* could be clinically very useful: following the introduction and approval of second-generation anti-androgens, the percentage of patients with AR-negative tumours has increased from 11.7% (1996–2011) to 36.6% (2012–2016) [65]. Prognosis for AR-negative CRPC is poor with a median survival of 12 months for AR-negative small cell carcinoma [66]. A recent randomised-control trial found that the targeted PARPi olaparib (recently FDA approved for PCa in May 2020) increased progression-free survival in patients with mCRPC whose cancer progressed despite abiraterone or enzalutamide treatment [67]. This suggests, as with PARP, *BRD9* may be a good therapeutic target in patients who stop responding to second generation antiandrogens. Another study found *BRD9* depletion increased cancer cell chemotherapy sensitivity as well as sensitivity to olaparib in Ovarian cancer, suggesting a drug targeting *BRD9* may work synergistically with olaparib [60].

### 4.3. Co-Targeting BRD9

The literature suggests that suggest BRD9 interacts with other genes in the SWI/SNF complex and cooperates with BET proteins [18,19].

We provide evidence showing that *BRD9* correlates and associates with many genes in the SWI/SNF complex as well as most genes in the BET complex. BRD9 could therefore be a viable co-target alongside other genes in these complexes. Though their positive correlation is very weak, *GLTSCR1* and its paralog *GLTSCR1-like* (*GLTSCR1-L*), the other unique GBAF complex components, interact with *BRD9* [18]. Interestingly, *BRD9* appears to be weakly positively correlated with *SMARCB1*, despite these genes being intrinsic competitors during SWI/SNF complex formation [34]. Alpsoy et al. showed that BET complex and *BRD9* inhibition exhibit similar effects, and that *BRD9* interacts with *BRD2* and *BRD4*, as well as with *BRD9* and BET proteins, regulating similar gene sets [19]. Though the authors discussed the effects of targeting the GBAF subunit, they did not investigate other SWI/SNF subunits and *BRD9*’s interactions with these.

The literature suggests that *BRD9* may interact with genes in the JAK-STAT, MAPK, and PI3K-AKT-mTor pathways. It has been shown that BRD9 depletion (at the mRNA and protein level) upregulates *SOCS3*, which, in turn, inhibits *STAT5*, reducing JAK-STAT pathway activation in acute myeloid leukaemia. Though *BRD9* is not correlated with *SOCS3* at a significant level in most cohorts and has a weakly positive correlation overall, *BRD9* and *STAT5A/B* mRNA expression are positively correlated [68]. *BRD9* is notably correlated with *AKT1* and *AKT 2* as well as other downstream effector protein genes including *RPTOR*, *MLST8*, *PRR5*, and *MAPKAP1*) in the PI3K-AKT-mTOR pathway and associates with a few genes in this signalling cascade. Though there is no uniform *BRD9*-*MYC* correlation, one study found that BRD9 inhibition reduced *MYC* levels in breast epithelial cell lines [69]. It has also been shown that MiR-140-3p (a tumour-suppressive miRNA) targets *BRD9* mRNA, downregulating *MYC* and reducing proliferation [35]. Interestingly, *BRD9* is negatively correlated with *PIK3CA*; its correlation with *KRAS* is non-uniform, though it has been shown that BRD9 may mediate a *PI3KCA*-*KRAS* mutant oncogenic cancer phenotype [69]. *BRD9* positively correlates with most genes in the MAPK pathway with a few exceptions and has some associations with this signalling cascade. Given its correlations with genes in this pathway and trametinib being trialled here in PCa, BRD9 has most promise as a co-target with this pathway.

We provide some evidence (in two treatment-naïve cohorts) that *BRD9* is overexpressed in *ERG* fusion positive cancers and that ERG expression is very weakly correlated with *BRD9* expression. This suggests that BRD9 may be a worthwhile co-target alongside *ERG*. Our data show that *BRD9* mutations are less prevalent in *ERG* fusion positive cancers, in line with the literature suggesting these cancers are less likely to have mutated SWI/SNF complexes [70,71]. BRD9 warrants further investigation in *ERG* fusion positive cancers and possibly co-targeted with *ERG* inhibition, as *ERG* depends on BAF complexes for its epithelial to mesenchymal transition (EMT) [72].

### 4.4. Limitations

This work investigated publicly available cohorts, which have limitations including that not all cohorts had mutation data. Some cohorts had small sample sizes and most lacked data on patients with more advanced stage cancers. Most cohorts also did not include information on treatment response, meaning it is difficult to conclude whether BRD9 has potential as a predictive biomarker using publicly available data. In some cohorts, original papers were unclear as to whether ‘normal’ or ‘benign’ samples were adjacent tissue from prostate cancer patients, or healthy tissue from patients who were determined not to have prostate cancer. This can lead to uncertainty in analysis. Some of the less recent data also uses grading classifications that we would not now, e.g., Gleason Grade 5 is now not considered cancer [72].

Another limitation is that, in each dataset, some samples are lost from each analysis due to the lack of data. For example, of the 491 patients in the TCGA cohort, there were only data for primary therapy outcome for 223 of these patients (45%), and there were only data for 284 (58%) patients on whether they were indicated for postoperative targeted radiotherapy.

We were also unable to investigate protein expression, as publicly available data regarding BRD9 protein expression were unavailable, which was a limitation of this work, where all the expression data were regarding mRNA.

In addition, it is not possible to determine causation—we can only study correlation in publicly available datasets; in the future, overexpression and knockdown studies must be performed to establish causation.

## 5. Conclusions

This approach to characterising *BRD9* using cancer omics cohorts has yielded promising results and, alongside the one published paper, forms a basis for future study of BRD9 in PCa. *BRD9* has most potential as a diagnostic biomarker and drug target in metastatic disease. Given its overexpression in patients with PCa across the datasets investigated, BRD9 could form part of a future PCa biomarker panel. This work warrants further experimental work, in vitro, in vivo, and ex vivo, to continue determining where and how targeting BRD9 could be successful in PCa.

## Figures and Tables

**Figure 1 biomolecules-11-01794-f001:**
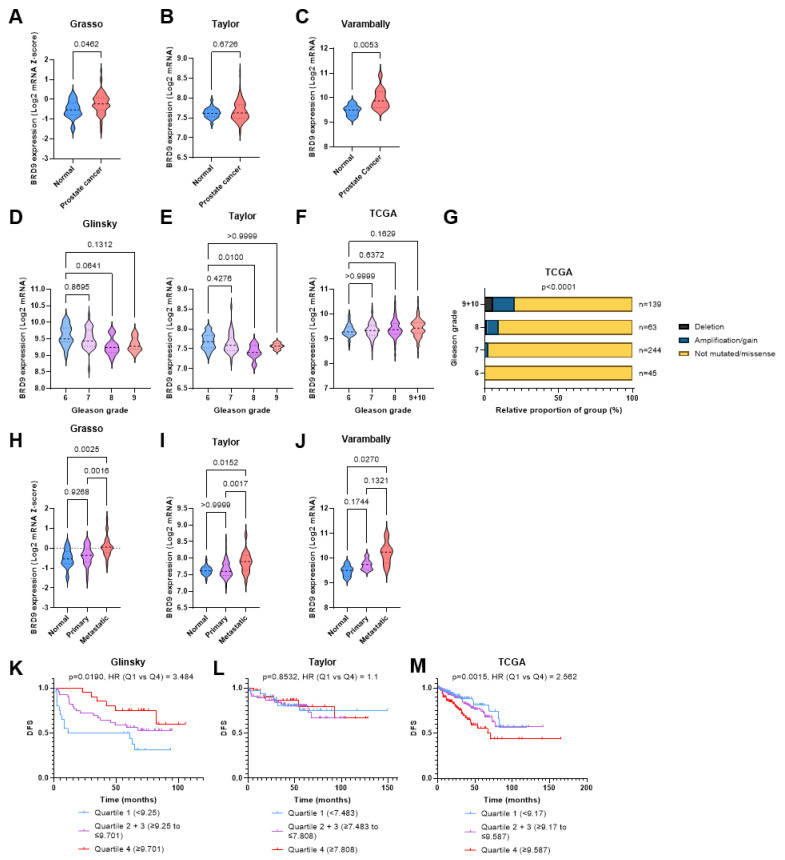
BRD9 as a diagnostic and prognostic biomarker in cancer omics cohorts. (**A**) Violin plot showing BRD9 expression in normal and PCa patients in the Grasso cohort. *p*-value was obtained using Welch’s *t*-test. (**B**) Violin plot showing BRD9 expression in normal and PCa patients in the Taylor cohort. *p*-value was obtained using Mann-Whitney U test. (**C**) Violin plot showing BRD9 expression in normal and PCa patients in the Varambally cohort. *p*-value was obtained using Welch’s *t*-test. (**D**) Violin plot showing how BRD9 expression varies with Gleason grade in the Glinsky cohort. *p*-values were obtained using Dunnett’s T3 post-hoc test. (**E**) Violin plot showing how BRD9 expression varies with Gleason grade in the Taylor cohort. *p*-values were obtained using Dunn’s post-hoc test. (**F**) Violin plot showing how BRD9 expression varies with Gleason grade in the TCGA cohort. *p*-values were obtained using Dunnett’s T3 post-hoc test. (**G**) Stacked bar chart showing BRD9 mutation distribution at different Gleason grades and the number of patients at each grade in the TCGA cohort. *p*-value was obtained via Fisher’s exact test using the Monte Carlo simulation. Grades 9 and 10 have been combined due to an n = 4 sample size at grade 10. (**H**) Violin plot showing how BRD9 expression varies with cancer progression in the Grasso cohort. *p*-values were obtained using Dunn’s post-hoc test. (**I**) Violin plot showing how BRD9 expression varies with cancer progression in the Taylor cohort. *p*-values were obtained using Dunn’s post-hoc test. (**J**) Violin plot showing how BRD9 expression varies with cancer progression in the Varambally cohort. *p*-values were obtained using Dunnett’s T3 post-hoc test. (**K**) DFS shown via KM survival curve in the Glinsky cohort. *p* value was obtained from a Mantel-Cox test and HR was calculated from a cox regression model by comparing the highest and lowest quartiles of BRD9 expression. (**L**) DFS shown via KM survival curve in the Taylor cohort. *p* value was obtained from a Mantel-Cox test and HR was calculated from a cox regression model by comparing the highest and lowest quartiles of BRD9 expression. (**M**) DFS shown via KM survival curve in the TCGA cohort. *p* value was obtained from a Mantel-Cox test and HR was calculated from a cox regression model by comparing the highest and lowest quartiles of BRD9 expression. Gleason grade (1–5) increases with more aggressive and less well-differentiated cancer. Overall grade (6–10) is the sum score of the two most prevalent grades in the sample. Y axis scales on violin plots vary due to experimental variation.

**Figure 2 biomolecules-11-01794-f002:**
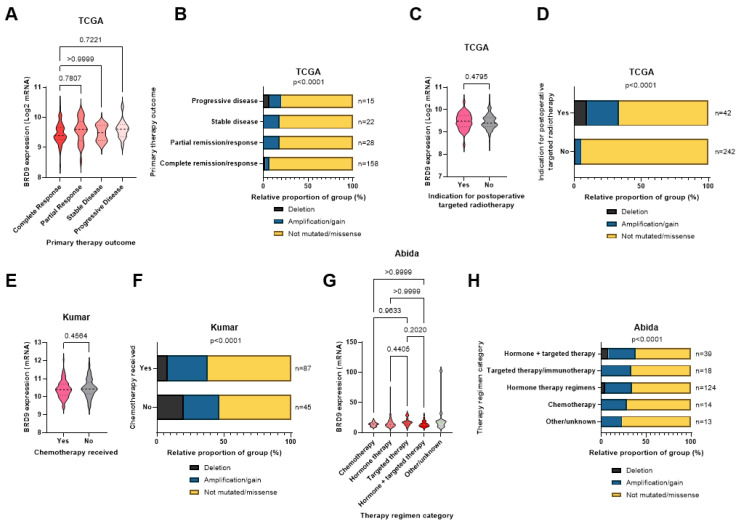
*BRD9* as a predictive biomarker in PCa in cancer omics cohorts. (**A**) Violin plot showing how *BRD9* expression varies with primary therapy outcome in the TCGA cohort. *p*-values were obtained using Dunn’s post-hoc test. (**B**) Stacked bar chart showing *BRD9* mutation distribution across primary therapy outcomes and the number of patients in each sample in the TCGA cohort. *p*-value was obtained via Fisher’s exact test using the Monte Carlo simulation. (**C**) Violin plot showing *BRD9* expression in patients indicated for adjuvant radiotherapy (left) and those who were not (right) in the TCGA cohort. *p*-value was obtained using a Mann Whitney U test. (**D**) Stacked bar chart showing *BRD9* mutation distribution in patients who were and were not indicated for adjuvant radiotherapy and the number of patients in each sample in the TCGA cohort. *p*-value was obtained using Fisher’s exact test. (**E**) Violin plot showing *BRD9* expression in patients who received chemotherapy (left) and those who did not (right) in the Kumar cohort. *p*-value was obtained using a Mann Whitney U test. (**F**) Stacked bar chart showing *BRD9* mutation distribution in patients who did and did not receive chemotherapy and the number of patients in each sample in the Kumar cohort. *p*-value was obtained via Fisher’s exact test using the Monte Carlo simulation. (**G**) Violin plot showing how *BRD9* expression varies with therapy regimen in the Abida cohort. *p*-values were obtained using Dunn’s post-hoc test. (**H**) Stacked bar chart showing *BRD9* mutation distribution in patients across therapy regimens the number of patients in each sample in the TCGA cohort. *p*-value was obtained using Fisher’s exact test. Regimen categories were combined as above due to small sample sizes (n) of various individual regimens. Y axis scales on violin plots vary due to experimental variation.

**Figure 3 biomolecules-11-01794-f003:**
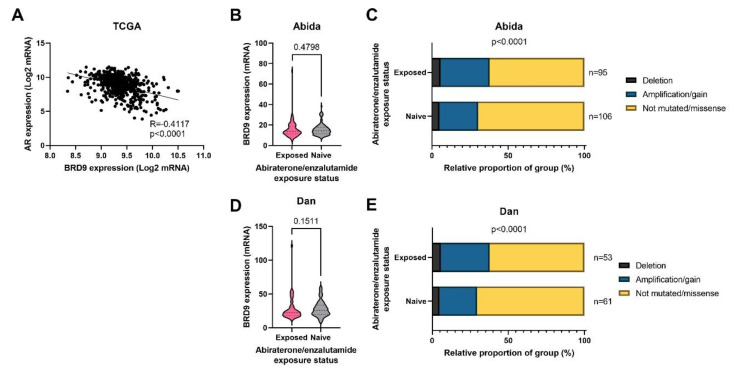
*BRD9* as a potential drug target in CRPC in cancer omics cohorts. (**A**) Scatterplot with line of best fit showing the correlation between *BRD9* and *AR* expression in the TCGA cohort. Spearman’s R and its associated two-tailed *p*-value have been calculated. (**B**) Violin plot showing *BRD9* expression in who patients were and were not treated with the second generation antiandrogens abiraterone and enzalutamide in the Abida cohort. *p*-value was obtained using a Mann Whitney U test. (**C**) Stacked bar chart showing *BRD9* mutation distribution in patients who were and were not treated with the second generation antiandrogens abiraterone and enzalutamide and the number of patients in each sample in the Abida cohort. *p*-value was obtained using Fisher’s exact test. (**D**) Violin plot showing *BRD9* expression in who patients were and were not treated with the second generation antiandrogens abiraterone and enzalutamide in the Dan cohort. *p*-value was obtained using a Mann Whitney U test. (**E**) Stacked bar chart showing *BRD9* mutation distribution in patients who were and were not treated with the second generation antiandrogens abiraterone and enzalutamide and the number of patients in each sample in the Dan cohort. *p*-value was obtained using Fisher’s exact test. Y axis scales on violin plots vary due to experimental variation.

**Figure 4 biomolecules-11-01794-f004:**
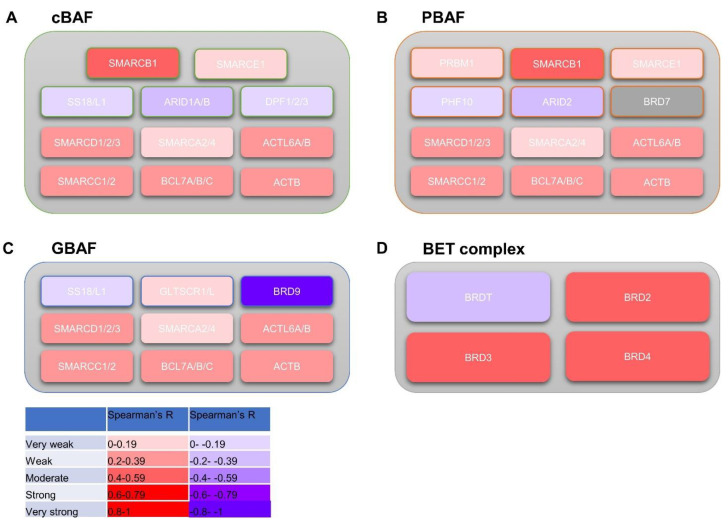
An overview of *BRD9* correlation with genes composing the SWI/SNF and BET complexes. Heatmap diagrams showing mean significant Spearman’s R *BRD9* correlation from datasets with gene subunits composing SWI/SNF subcomplexes (cBAF, PBAF, GBAF) and BET complex. (**A**) Heatmap diagram showing mean significant Spearman’s R *BRD9* correlation from datasets with gene subunits composing the cBAF subcomplex. (**B**) Heatmap diagram showing mean significant Spearman’s R *BRD9* correlation from datasets with gene subunits composing the PBAF subcomplex. (**C**) Heatmap diagram showing mean significant Spearman’s R *BRD9* correlation from datasets with gene subunits composing the GBAF subcomplex. SWI/SNF diagrams adapted from Centore et al.’s Figure 1 [15]. Image created using Microsoft PowerPoint. (**D**) Heatmap diagram showing mean significant Spearman’s R *BRD9* correlation from datasets with gene subunits composing the BET complex.

**Figure 5 biomolecules-11-01794-f005:**
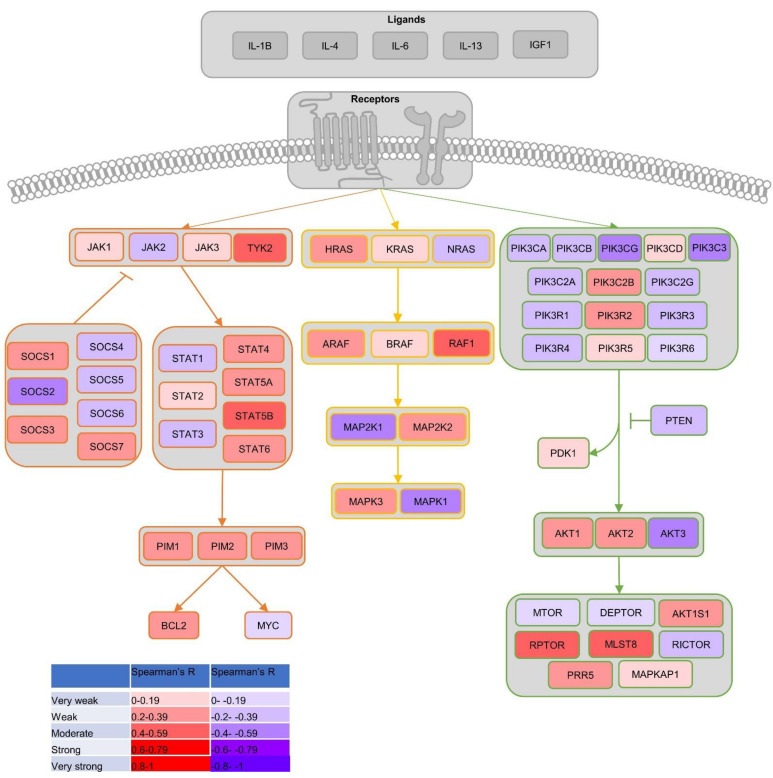
An overview of *BRD9* correlation with common signalling pathways involved in PCa. Flowchart signalling pathway diagram/heatmap showing mean significant Spearman’s R *BRD9* correlation from datasets with genes in the JAK-STAT, MAPK and PI3K-AKT-mTOR signalling pathways. Adapted and expanded from Luzczak et al.’s Figure 2 [57]. Image created using Microsoft PowerPoint.

**Figure 6 biomolecules-11-01794-f006:**
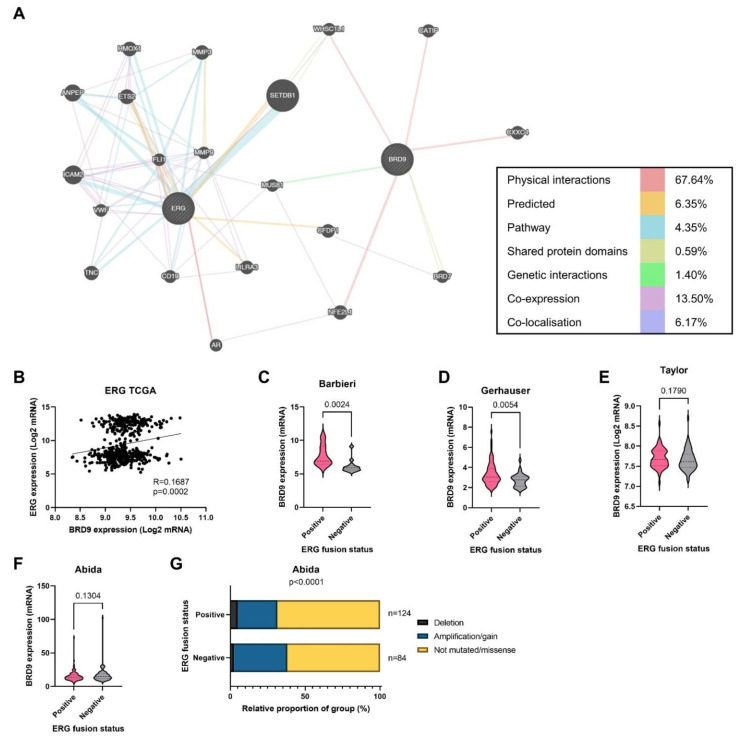
The potential of ERG as a co-target with *BRD9*. (**A**) Gene network map showing the associations between *BRD9* and *ERG*. (**B**) Scatterplot with line of best fit showing the correlation between *BRD9* and *ERG* expression in the TCGA cohort. (**C**) Violin plot showing *BRD9* expression in who patients were *ERG* (*TMPRSS2-ETS* fusion) positive (right) and negative (left) in the Barbieri cohort. *p*-value was obtained using a Mann Whitney U test. (**D**) Violin plot showing *BRD9* expression in who patients were ERG positive (right) and negative (left) in the Gerhauser cohort. *p*-value was obtained using a Mann Whitney U test. (**E**) Violin plot showing *BRD9* expression in who patients were *ERG* positive (right) and negative (left) in the Taylor cohort. *p*-value was obtained using a Mann Whitney U test. (**F**) Violin plot showing *BRD9* expression in who patients were *ERG* positive (right) and negative (left) in the Abida cohort. *p*-value was obtained using a Mann Whitney U test. (**G**) Stacked bar chart showing *BRD9* mutation distribution in patients who were *ERG* positive and negative and the number of patients in each sample in the Abida cohort. *p*-value was obtained using Fisher’s exact test. Y axis scales on violin plots vary due to experimental variation. Gene network map created using https://genemania.org/. (accessed on 19 April 2021).

**Table 1 biomolecules-11-01794-t001:** Characteristics of human prostate cancer omics datasets investigated. Table summarising characteristics of publicly available cancer omics datasets and their patients. For each clinical attribute investigated, there was not necessarily data for each patient in the cohort.

	TCGA [43]	Glinsky [44]	Grasso [45]	Taylor [46]	Varambally [47]	Gerhauser [48]	Barbieri [49]	Ren [50]	Abida [51]	Dan [52]	Kumar [53]
**Cancertool name**	TCGA	Glinsky	Grasso	Taylor	Varambally	N/A	N/A	N/A	N/A	N/A	N/A
**cBioPortal name**	TCGA Firehose Legacy	N/A	Metastatic Prostate Adenocarcinoma (MCTP, Nature 2012)	Prostate Adenocarcinoma (MSKCC, Cancer Cell 2010)	N/A	Prostate Cancer (DKFZ, Cancer Cell 2018)	Prostate Adenocarcinoma (Broad/Cornell, Nat Genet 2012)	Prostate Adenocarcinoma (SMMU, Eur Urol 2017)	Metastatic Prostate Adenocarcinoma (SU2C/PCF Dream Team, PNAS 2019)	Metastatic Prostate Cancer SU2C/PCF Dream Team, Cell 2015)	Prostate Adenocarcinoma (Fred Hutchinson CRC, Nat Med 2016)
**Number of patients (N)**	499	79	61	218	13	251	112	65	429	150	176
**Cancer type**	Primary prostate adenocarcinoma	Recurrent (n = 37) and nonrecurrent (n = 42) disease	Heavily penetrated CRPC (n = 50), high grade localised PCa (n = 11)	Primary tumours (n = 81), metastatic disease (n = 37)	Primary tumours (n = 7), metastatic disease (n = 6)	Primary prostate adenocarcinoma	Prostate adenocarcinoma	Prostate adenocarcinoma	Metastatic CRPC	Metastatic CRPC	Primary and metastatic—mCRPC (n = 63)
**Treatment (prior to specimen collection)**	Naïve	Undergoing routine treatment	Prostatectomy, chemotherapy, hormone therapy, radiotherapy, palliative radiotherapy. Localised cancers were treatment naïve.	No information	No information	Treatment naïve (except for two patients receiving neoadjuvant hormone therapy)	Treatment-naïve	Treatment-naïve	Standard of care (including second generation antiandrogens)/enrolled in a clinical trial (e.g., targeted therapy)	Various—second generation antiandrogens, clinical trial, taxane chemotherapy	ADT followed by second-generation antiandrogens (following disease progression) and docetaxel chemotherapy
**Method of sample collection**	Radical prostatectomy	During therapeutic/diagnostic procedures	Rapid autopsy (CRPC) and radical prostatectomy (localised)	Radical prostatectomy	Radical prostatectomy and rapid autopsy	Radical prostatectomy	Radical prostatectomy	Radical prostatectomy	Radiographic-guided biopsy	Radiographic-guided biopsy	Rapid autopsy
**Gleason grade**	5: -	5: -	5: -	5: 2 (1%)	-	5: -	5: -	5: -	5: -	-	-
	6: 45 (9%)	6: 15 (19%)	6: -	6: 101 (47%)	-	6: 13 (11%)	6: 4 (20%)	6: 5 (8%)	6: 15 (9%)	-	-
	7: 244 (50%)	7: 45 (57%)	7: 2 (18%)	7: 77 (36%)	-	7: 86 (74%)	7: 13 (65%) 7.5: 1 (5%)	7: 37 (57%)	7: 49 (29%)	-	-
	8: 63 (13%)	8: 10 (13%)	8: 4 (36%)	8: 19 (9%)	-	8: 1 (1%)	8: 2 (10%)	8: 9 (14%)	8: 23 (14%)	-	-
	9: 135 (27%)	9: 9 (11%)	9: 5 (46%)	9: 15 (7%)	-	9: 15 (13%)	9: -	9: 12 (18%)	9: 67 (40%)	-	-
	10: 4 (1%)	10: -	10: -	10: -	-	10: 1 (1%	10: -	10: 2 (3%)	10: 13 (8%)	-	-
	11: -	11: -	11: -	11: -	-	11: -	11: -	11: -	11: 1 (1%)	-	-
**Total patients (n) for whom a Gleason grade was available**	491 (100%)	79 (100%)	11 (100%) (High grade localised PCas only)	214 (100%)	No data available	116 (100%)	20 (100%)	65 (100%)	168 (101%)	No data available	No data available

**Table 2 biomolecules-11-01794-t002:** Classification of correlation strength. Table summarising correlation strength.

	Spearman’s R (Positive)	Spearman’s R (Negative)
Very weak	0–0.19	0–−0.19
Weak	0.2–0.39	−0.2–−0.39
Moderate	0.4–0.59	−0.4–−0.59
Strong	0.6–0.79	−0.6–−0.79
Very strong	0.8–1	−0.8–−1

## Data Availability

Data available in a publicly accessible repository at: Cancertool DOI: 10.1158/0008-5472.CAN-18-1669 [39]; cBioPortal DOI: 10.1158/2159-8290.CD-12-0095 and DOI: 10.1126/scisignal.2004088 [40,41]; Genemania DOI: 10.1126/scisignal.2004088 [42].

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
