# Peer review of "Exploring the Value of BRD9 as a Biomarker, Therapeutic Target and Co-Target in Prostate Cancer"

_biomolecules, 2021, doi:10.3390/biom11121794_

Round 1

Reviewer 1 Report

Title:

The title reflects the aims of the study well.

Abstract:

The abstract reflects their work and the aims.

Introduction:

The introduction reflects their work in a plausible way.

There are some points that I would suggest.

Please change ‘SWI/SNF and BET complexes’ to ‘The relationship between BRD9 and SWI/SNF and BET complexes’.

Also, as mentioned in the Discussions section, I suggest it would be better to include the reference 30, 64 and explain it in more detail.

Materials and Methods:

The materials and methods were clearly presented and could be followed easily in figure and tables.

Results:

The results were clearly presented and could be followed easily in figure and tables.

Discussions:

The discussions reflect their work in a plausible way.

There are some points that I would suggest.

In line 233-234, Is it correct?

Conclusions:

The conclusion reflects their work in a plausible way.

Author Response

Reviewer 1 Comments & Author Response 

Title:

The title reflects the aims of the study well.

The authors thank Reviewer 1 for their comment.

Abstract:

The abstract reflects their work and the aims.

The authors thank Reviewer 1 for their comment.

Introduction:

The introduction reflects their work in a plausible way.

The authors thank Reviewer 1 for their comment.

There are some points that I would suggest.

Please change ‘SWI/SNF and BET complexes’ to ‘The relationship between BRD9 and SWI/SNF and BET complexes’.

The authors thank Reviewer 1 for their comment and have amended the introduction as suggested.

Also, as mentioned in the Discussions section, I suggest it would be better to include the reference 30, 64 and explain it in more detail.

The authors thank Reviewer 1 for their comment and have amended the introduction to reflect this.

Materials and Methods:

The materials and methods were clearly presented and could be followed easily in figure and tables.

The authors thank Reviewer 1 for their comment.

Results:

The results were clearly presented and could be followed easily in figure and tables.

The authors thank Reviewer 1 for their comment.

Discussions:

The discussions reflect their work in a plausible way.

The authors thank Reviewer 1 for their comment.

There are some points that I would suggest.

The authors cannot find these comments. There is no additional text or file available within the system. Could the reviewer or editor advise where these comments are please?

In line 233-234, Is it correct?

These lines read ‘AR expression is moderately negatively correlated with AR expression in the TCGA cohort.’ The authors can confirm that an R value of -0.4117, with p<0.0001 is considered a moderate negative correlation. If the line count has moved during the review process, perhaps the reviewer was referring to another line?

Conclusions:

The conclusion reflects their work in a plausible way.

The authors thank Reviewer 1 for their comments, their expertise, and efficiency.

Reviewer 2 Report

There are several minor comments such as:

  • PCa is not  always abreviated.
  • A summary abstract will improve the text.
  • Tables should be named at the beginning not at the end.
  • Sometimes you use alpha or p, for indicating p value. Please use always the same.
  • It is strange to find a text with this title "It is unclear whether BRD9 could play a role as a therapeutic target in CRPC."
  • Latin words like "in vivo" and similars, should be in cursive.

Also some major comments such as:

  • In introduction section a more robust information in why you choose BRD9 to develop the study will help the reader.
  • You also mentioned that there is only one paper published, indeed there are two now in pubmed.
  • Materials and methods: before starting with cancertool you should summarize how do you plan your study, it is not clear how you develop it.
  • Clinical correlations and co-expression analysis, in this section, what clinical data do you have? you do not mention it.
  • In Comparing mutation distribution  section, it is the same. What mutations do you look for? somatic? or what type? could you list them?
  • Figures 1 and 2 in results section is too small.
  • Limitations did not explain percentage of samples you loose in each analysis due to the lack of data.
  • Conclusion is to short and not enough motivating.

Author Response

Reviewer 2 Comments & Author Response

There are several minor comments such as:

PCa is not  always abreviated.

The authors thank Reviewer 2 for their comment and have updated the main body of the text to reflect this. It has not been abbreviated for clarity when another abbreviation is being defined.

A summary abstract will improve the text.

The authors have made changes to the abstract as requested by other reviewers and are unclear as to whether this reviewer would also like a separate shorter abstract. Could the reviewers (or editor) please clarify?

Tables should be named at the beginning not at the end.

The authors thank Reviewer 2 for their comment and have included table names at the top of each table.

Sometimes you use alpha or p, for indicating p value. Please use always the same.

The authors thank Reviewer 2 for their comment and have amended the results section accordingly.

It is strange to find a text with this title "It is unclear whether BRD9 could play a role as a therapeutic target in CRPC."

The authors thank Reviewer 2 for their comment and have changed the aforementioned heading to ‘BRD9 may play a role as a therapeutic target in CRPC’.

Latin words like "in vivo" and similars, should be in cursive.

The authors did not use italics for such phrases, as per section 3.6 of the MDPI style guide ‘Foreign words do not need to be highlighted or italicized, including Latin terms such as ‘in situ’. ‘

Also some major comments such as:

In introduction section a more robust information in why you choose BRD9 to develop the study will help the reader.

The authors thank Reviewer 2 for their comment and have amended the article accordingly.

You also mentioned that there is only one paper published, indeed there are two now in pubmed.

The authors thank Reviewer 2 for their comment and have added a reference to Alpsoy et al’s ‘Glioma tumor suppressor candidate region gene 1 (GLTSCR1) and its paralog GLTSCR1-like form SWI/SNF chromatin remodeling subcomplexes.’, however this is not about BRD9 in prostate cancer, so the authors feel this is tangentially relevant.

Materials and methods: before starting with cancertool you should summarize how do you plan your study, it is not clear how you develop it.

The authors thank Reviewer 2 for their comment and have added the following sentence ‘Cancertool, cBioPortal and Genemania were used to access data to investigate BRD9 expression, how BRD9 expression correlated with that of other genes and in different types of patient as well as how BRD9 may interact with other genes.’ To clarify how we planned our study.

Clinical correlations and co-expression analysis, in this section, what clinical data do you have? you do not mention it.

The authors thank Reviewer 2 for their comment. The datasets accessed are summarised in table 1 and these contain the data access. The heading ahs been changed to ‘Gene correlations and co-expression analysis’, to more accurately reflect the analysis carried out here.

In Comparing mutation distribution  section, it is the same. What mutations do you look for? somatic? or what type? could you list them?

The authors thank Reviewer 2 for their comment and have amended the text to read ‘Where somatic mutation data (amplification, gain of function, missense, shallow deletion and deep deletions) was available’ to clarify details regarding mutations looked at.

Figures 1 and 2 in results section is too small.

The authors thank Reviewer 2 for their comment. As these figures have been reformatted, they ar smaller than the originals. The authors are happy to rethink these if the editors agree?

Limitations did not explain percentage of samples you loose in each analysis due to the lack of data.

The authors thank Reviewer 2 for their comment and agree that this is a limitation of the work done. The authors have added the following sentence to the limitations section to reflect this; ‘Another limitation is that in each dataset, some samples are lost from each analysis due to the lack of data. For example, of the 491 patients in the TCGA cohort, there was only data for primary therapy outcome for 223 of these patients (45%) and there was only data for 284 (58%) patients on whether they were indicated for postoperative targeted radiotherapy.’

Conclusion is to short and not enough motivating.

The authors than Reviewer 2 for their comments and expertise and have amended the conclusion accordingly.

Reviewer 3 Report

Although this is solely a bioinformatic study, it may be of interest to other researchers.

A number of issues must be resolved:
Figure 4 needs further clarification:
Figure 4B - colour code for BRD7 is not explained
Figure 4C - BRD9 negatively correlates with BRD9? This is confusing.
Figure 4D - figure legend is missing

Incorrect statements
- "When comparing BRD9 expression in patients, available data, normal patients are not actually healthy  patients, rather they have benign prostates."
- "Gleason Grade 5 is now not considered cancer"

Arguable statements:
"Our findings show BRD9 is overexpressed in PCa and therefore may play a role as a  diagnostic biomarker in  this disease." - Standard PSA screening is well-established in PCa diagnosis, as well as other novel methods, such as MRI or PHI. Clinical problem is overdiagnosis of indolent carcinomas

The following statement is exaggerated: "While the cell line data is promising, cell lines cannot create a biomimetic environment, and are often contaminated or subject to genetic drift" - Although the cross-contamination of cell lines might be a problem, good laboratories and journals request STR profiling for authentication. 

"a preferable experiment would be to compare BRD9 expression with response to antiandrogens in all patients –  data which is not currently available within these datasets." - This is obvious, but unrealistic to expect.

There is no "Smith" in reference 30: "Smith’s paper on BRD9 in PCa suggests BRD9 could play a role in CRPC with BRD9 knockdown reducing the viability of AR-positive cell lines (including castration-resistant cell lines)[30]."

The following sentence in the Discussion does not reflect Figure 5: "BRD9 correlates with genes in the PI3K-AKT-mTOR pathway and associates with a few genes in this signaling cascade"

Olaparib is not a target: "This suggests, like Olaparib, BRD9 may be a good therapeutic target in patients who stop responding to second generation antiandrogens." Lower cases should be used for olaparib and ovarian cancer: "sensitivity to Olaparib in Ovarian cancer"

Proofreading is needed:
- "AR expression is moderately negatively correlated with AR expression in the TCGA cohort."
- "it is difficult conclude"
- "Mi-140-3p"

Author Response

Reviewer 3 Comments & Author Response 

Although this is solely a bioinformatic study, it may be of interest to other researchers.

A number of issues must be resolved:
Figure 4 needs further clarification:
Figure 4B – colour code for BRD7 is not explained
Figure 4C – BRD9 negatively correlates with BRD9? This is confusing.
Figure 4D – figure legend is missing

The authors thank Reviewer 3 for their comments. BRD7 is in grey as there were no significant correlations between BRD7 and BRD9 were non-significant. This has been added to the figure legend for clarification. Regarding figure 4C, this was a mistake and has been corrected. Thank you for bringing this to our attention. Regarding figure 4D, the authors thank reviewer 4 for binging this to our attention and have now corrected this.

Incorrect statements
- “When comparing BRD9 expression in patients, available data, normal patients are not actually healthy  patients, rather they have benign prostates.”
- “Gleason Grade 5 is now not considered cancer”

The authors thank Reviewer 3 for their comments. The first statement mentioned here has been rephrased to read ‘In some cohorts, original papers were unclear as to whether ‘normal’ or ‘benign’ samples were adjacent tissue from prostate cancer patients, or healthy tissue from patients who were determined not to have prostate cancer. This can lead to uncertainty in analysis’ and hope this provides some clarity. Regarding the second statement, a reference has been added to provide evidence for this statement, that historically a Gleason sum score of 5 was considered cancer.

Arguable statements:
“Our findings show BRD9 is overexpressed in Pca and therefore may play a role as a  diagnostic biomarker in  this disease.” – Standard PSA screening is well-established in Pca diagnosis, as well as other novel methods, such as MRI or PHI. Clinical problem is overdiagnosis of indolent carcinomas

The authors thank Reviewer 3 for their comment and have added that While BRD9 could represent a good diagnostic biomarker in Pca, it may not necessarily mean that this is needed, with PSA being widely used. Biomarkers that more accurately predict aggressive disease are more needed,  so we went on to identify whether BRD9 could represent a more clinically useful biomarker here.

The following statement is exaggerated: “While the cell line data is promising, cell lines cannot create a biomimetic environment, and are often contaminated or subject to genetic drift” – Although the cross-contamination of cell lines might be a problem, good laboratories and journals request STR profiling for authentication. 

The authors thank Reviewer 3 for their comment and have amended this sentence slightly and ensured it is backed up with references.

“a preferable experiment would be to compare BRD9 expression with response to antiandrogens in all patients –  data which is not currently available within these datasets.” – This is obvious, but unrealistic to expect.

The authors thank Reviewer 3 for their comment and agree this is unrealistic to expect and have amended the sentence.

There is no “Smith” in reference 30: “Smith’s paper on BRD9 in Pca suggests BRD9 could play a role in CRPC with BRD9 knockdown reducing the viability of AR-positive cell lines (including castration-resistant cell lines)[30].”

The authors thank Reviewer 3 for their comment and bringing this to our attention – this was a mistake and has been corrected.

The following sentence in the Discussion does not reflect Figure 5: “BRD9 correlates with genes in the PI3K-AKT-mTOR pathway and associates with a few genes in this signaling cascade”

The authors thank Reviewer 3 for their comment and have rephrased the sentence to reflect the figure.

Olaparib is not a target: “This suggests, like Olaparib, BRD9 may be a good therapeutic target in patients who stop responding to second generation antiandrogens.” Lower cases should be used for laparib and ovarian cancer: “sensitivity to Olaparib in Ovarian cancer”

The authors thank Reviewer 3 for their comment and have amended the manuscript accordingly.

Proofreading is needed:
- “AR expression is moderately negatively correlated with AR expression in the TCGA cohort.”
- “it is difficult conclude”
- “Mi-140-3p”

The authors thank Reviewer 3 for their expertise and feedback and have proofread the manuscript and made the suggested corrections.

Round 2

Reviewer 2 Report

  • In the title  BRD9  is not well written.
  • BRD9 is not always in cursive as well as other genes abbreviated.
  • Table 1 is in bad quality.
  • Introduction section still being uncomplete according to expose BRD9 in PCa and other tumors.

Author Response

In the title  BRD9  is not well written.

Thank you, we have fixed this

BRD9 is not always in cursive as well as other genes abbreviated.

This was edited during the review process in response to others, to be in italics when referring to the gene, and not in italics when referring to the protein. For example in the paragraph below we discuss gene knockdown and protein interaction:

“Though it has shown promise as a biomarker and therapeutic target in other cancers, BRD9 has not been well studied in PCa to date with only one paper published in December 2020 [32-34][19]. This paper showed that BRD9 inhibition and knockdown have overlapping effects, reducing AR-positive cell line growth both in vivo and in vitro. The authors also found that BRD9 interacts with the AR in PCa cell lines, even those resistant to androgen deprivation and inhibition. They also provided cell-line evidence that BRD9 and the BET complex associate with each other and interact given they have overlapping transcriptional targets.”

Table 1 is in bad quality.

Thanks, we agree there is low resolution here. We have now separately submitted original files for all figures and tables so that all can be in higher resolution.

Introduction section still being uncomplete according to expose BRD9 in PCa and other tumors.

Thanks, we have added further information around BRD9 in other cancers as well as PCa to the introduction.

Reviewer 3 Report

The manuscript has been improved, however, minor proofreading is still needed - there are two chapters "Cancertool, cBioPortal and Genemania" in the Methods section.

Author Response

The manuscript has been improved, however, minor proofreading is still needed - there are two chapters "Cancertool, cBioPortal and Genemania" in the Methods section.

Thanks, this has been fixed in the methods section, and further general proofreading has identified and fixed a small number of typographical errors.